# Fungal Gut Microbiome in Myasthenia Gravis: A Sub-Analysis of the MYBIOM Study

**DOI:** 10.3390/jof9050569

**Published:** 2023-05-13

**Authors:** Hedda Luise Verhasselt, Elakiya Ramakrishnan, Melina Schlag, Julian R Marchesi, Jan Buer, Christoph Kleinschnitz, Tim Hagenacker, Andreas Totzeck

**Affiliations:** 1Institute of Medical Microbiology, University Hospital Essen, University of Duisburg-Essen, D-45122 Essen, Germany; 2Department of Neurology and Center for Translational Neuro- and Behavioral Sciences (C-TNBS), University Hospital Essen, University of Duisburg-Essen, D-45122 Essen, Germany; 3Division of Digestive Diseases, Department of Metabolism, Digestion and Reproduction, Imperial College London, London SW7 2BX, UK

**Keywords:** myasthenia gravis, chronic inflammatory demyelinating polyradiculoneuropathy, ITS2, mycobiome

## Abstract

An altered gut microbiota is a possible contributing pathogenic factor in myasthenia gravis (MG), an autoimmune neuromuscular disease. However, the significance of the fungal microbiome is an understudied and neglected part of the intestinal microbiome in MG. We performed a sub-analysis of the MYBIOM study including faecal samples from patients with MG (*n* = 41), non-inflammatory neurological disorder (NIND, *n* = 18), chronic inflammatory demyelinating polyradiculoneuropathy (CIDP, *n* = 6) and healthy volunteers (n = 12) by sequencing the internal transcribed spacer 2 (ITS2). Fungal reads were obtained in 51 out of 77 samples. No differences were found in alpha-diversity indices computed between the MG, NIND, CIDP and HV groups, indicating an unaltered fungal diversity and structure. Overall, four mould species (*Penicillium aurantiogriseum, Mycosphaerella tassiana, Cladosporium ramonetellum* and *Alternaria betae-kenyensis*) and five yeast species (*Candida. albicans, Candida. sake, Candida. dubliniensis, Pichia deserticola* and *Kregervanrija delftensis*) were identified. Besides one MG patient with abundant *Ca. albicans*, no prominent dysbiosis in the MG group of the mycobiome was found. Not all fungal sequences within all groups were successfully assigned, so further sub-analysis was withdrawn, limiting robust conclusions.

## 1. Introduction

Myasthenia gravis (MG) is an autoimmune-mediated disorder with autoantibodies targeting different proteins of the neuromuscular junction [1]. Classification of different autoantibody types, particularly antibodies against the acetylcholine receptor (AChR), helped to clarify the mechanisms of symptoms such as exercise-induced muscle weakness. However, the pathogenic trigger of MG is yet not fully understood. Altered gut microbiota is a possible predisposing factor for MG [2,3]. Bacterial gut microbiota profiles of MG patients differ from those of healthy volunteers [3,4,5]. In the last few years, several cohort studies, as well as rat models, have implicated the bacterial gut microbiota in MG induction and severity [3,4,5,6]. However, sufficient data on fungal gut microbiota profiles of MG patients compared with those of patients with other inflammatory or non-inflammatory neurological diseases is still lacking. Specific compositions of the gut mycobiome are associated with autoimmune conditions such as multiple sclerosis (MS) [7,8], and an increase in *Candida* spp. is linked to neurological disorders such as autism spectrum disorders [9], schizophrenia [10] and Rett syndrome [11]. Although it is not fully understood how *C. albicans* contributes to the pathogenesis of these diseases, one well-established mechanism for the increase in *C. albicans* abundance seen in disease is that colonisation of *C. albicans* can drive T-helper-17 cell-mediated immune responses, leading to exacerbation [12]. Through the activation of pattern recognition receptors, mannans, fungal cell wall constituents from *C.* spp., induced Th17 and IL-23 responses, leading to the worsening of intestinal graft-versus-host disease and colitis in mice [13,14,15]. We present here a sub-analysis of the previous MYBIOM study, a single-centre observational study on bacterial gut microbiota with 77 participants, to determine whether the fungal gut microbiota is altered in MG patients compared with non-inflammatory neurological disorder (NIND), patients with chronic inflammatory demyelinating polyradiculoneuropathy (CIDP) and healthy volunteers (HV) without neurological disorders [3].

## 2. Materials and Methods

### 2.1. Study Design and Patients

MYBIOM was a single-centre observational study conducted at the Department of Neurology at the University Hospital of Essen in Germany from July 2017 to March 2018. The study design and the bacterial microbiome results have been published elsewhere [3]. In brief, MG patients were included based on their clinical characteristics, such as fluctuating fatigability and muscle weakness, a positive response to cholinesterase inhibitors and, optionally, a recorded decrease in repetitive motor nerve stimulation [16]. Three different groups of patients were recruited for comparison: subjects with NIND, those with CIDP diagnosis based on the European Federation of Neurological Societies/Peripheral Nerve Society diagnostic criteria [17] and HV without any underlying neurological or systemic inflammatory disease were recruited as controls. Study participants were 18 years or older and had no history of treatment with antibiotics within the previous 4 weeks, and no intentional consumption of probiotics or anti-obesity agents within 3 months prior to study participation. Exclusion criteria were chronic inflammatory bowel disease, short bowel syndrome, irritable bowel syndrome, pregnancy and recent treatment with chemotherapeutics or monoclonal antibodies.

### 2.2. Sample Collection, DNA Extraction and Sequencing 

Fresh faecal samples were collected from participants and were transported to the Institute of Medical Microbiology at 4°C within 12 h of specimen collection. A total of 77 faecal samples (MG (*n* = 41), CIDP (*n* = 6), NIND (*n* = 18) and HV (*n* = 12), Figure 1), as well as 6 negative controls (RNA-free water) and 2 positive controls (Escherichia coli and a mix of E. coli, Staphylococcus aureus and Corynebacterium striatum), were stored at −80°C until DNA extraction [3] Using a QIAmp Fast DNA Stool Mini kit (Qiagen). DNA aliquots were stored at −80°C until use. The internal transcribed spacer 2 (ITS2) region was amplified from faecal DNA using primers ITS3 and ITS4 (ITS3 F (5′- GCATCGATGAAGAACGCAGC-3′) and ITS4 R (5′- TCCTCCGCTTATTGATATGC-3′)) [18]. BaseClear B.V. (Leiden, the Netherlands) performed library preparation and sequencing on the Illumina MiSeq platform (10 k paired-end reads) as well as Illumina raw data processing. 

### 2.3. Illumina Demultiplexing

Paired-end (2 × 300 bp) sequence reads were generated using the Illumina MiSeq system. The sequences generated performed under accreditation according to the scope of BaseClear B.V. (L457; NEN-EN-ISO/IEC 17025). FASTQ read sequence files were generated using bcl2fastq version 2.20 (Illumina). Initial quality assessment was based on data passing the Illumina Chastity filtering. Subsequently, reads containing PhiX control signal were removed using an inhouse filtering protocol. In addition, reads containing (partial) adapters were clipped (up to a minimum read length of 50 bp). The second quality assessment was based on the remaining reads using the FASTQC quality control tool version 0.11.8. A total of 1,579,359 read-pairs were obtained from sequencing the ITS2 region after demultiplexing and filtering. Samples without fungal reads were excluded from further analysis. 

### 2.4. Processing and Statistical Analysis of Metataxonomic Data

Sequence processing was performed using DADA2 to cluster sequences into ASV and to provide taxonomic annotations. The DADA2 ITS Pipeline Workflow (1.8) was used to process the reads (https://benjjneb.github.io/dada2/ITS_workflow.html, accessed on 4 July 2022). Fungal taxonomy was assigned using the UNITE database (https://unite.ut.ee, accessed on 4 July 2022). Statistical analyses were performed in MicrobiomeAnalyst [19] and the data exported to CSVs for further analysis. Within-sample alpha-diversity indices [Chao1, Simpson, and Shannon] were calculated in the MicrobiomeAnalyst. For comparison between groups, the nonparametric Kruskal–Wallis and Mann–Whitney tests were calculated in GraphPad Prism v7.05 (GraphPad Software, San Diego, CA, USA, www.graphpad.com, accessed on 4 July 2022). Statistical tests with *p* ≤ 0.05 were considered significant.

## 3. Results

After exclusion of samples without fungal reads, 51 samples were further analysed ((MG (*n* = 27), CIDP (*n* = 4), NIND (*n* = 11) and HV (*n* = 9)). To assess differences in the gut microbiota of patients with MG, CIDP, NIND and HV, ecological features of the faecal fungal communities were evaluated using the alpha diversity indices Simpson’s, Shannon and Chao1. None of the indices were significantly different between groups (*p* > 0.05) (Figure 2).

The composition of fungal microbiota from faeces was characterized by the presence of yeasts and moulds. Subjects with failed assignment to fungal species belonged to all groups (CIDP 2/4, HV 7/9, MG 22/27 and NIND 8/11). 

Failed assignment affected either all fungal sequences per subject (CIDP 1/2, HV 4/7, MG 10/22 and NIND 3/8) or a portion. Overall, four mould species (*Penicillium aurantiogriseum, Mycosphaerella tassiana, Cladosporium ramonetellum* and *Alternaria betae-kenyensis*) and five yeast species (*C. albicans, C. sake, C. dubliniensis, Pichia deserticola* and *Kregervanrija delftensis*) were identified (Figure 3). In subject A07 (MG) *C. albicans* abundance dominated within the fungal community. Furthermore, no prominent dysbiosis of the mycobiome was found. Due to the limited numbers of successfully assigned fungal sequences, we refrained from further data analysis and representations. 

## 4. Discussion

In this study, extracted DNA from faecal samples from MYBIOM study participants was examined to assess the gut mycobiome. No difference was found comparing the alpha-diversity indices of the different study groups. Our findings include *Penicillium aurantiogriseum, Mycosphaerella tassiana, Cladosporium ramonetellum* and *Alternaria betae-kenyensis* among filamentous fungi. *Mycosphaerella tassiana*, the heterotypic synonym of *Cladosporium herbarum*, is a common fungus found worldwide, and its spores are highly prevalent in the air [20,21]. Together with *Cladosporium herbarum*, *Alternaria* spp. are common allergens, and the latter are known as major plant pathogens ubiquitously distributed in the environment. The genus *Alternaria* can also be found on normal human skin is part of the oral mycobiome in healthy individuals [22,23]. In vitro, *Alternaria* and *Cladosporium* species have been found to reveal acetylcholinesterase inhibitory activities [24,25]. Interestingly, acetylcholinesterase inhibitors are common therapeutics in MG, but further studies need to clarify potential benefits of these fungal genera in MG. As stool samples for this study were processed in a Class II safety cabinet, fungal spores possibly occurred during defecation and sampling or may have been swallowed by patients.

Among yeast, the species *Kregervanrija pseudodelftensis* and *Pichia deserticola* were found, both belonging to the Pichiaceae family, and together with *C. sake* they are associated with the consumption of fruits [26,27,28]. *C. albicans* causes serious infections in hospitalized patients, associated with high morbidity and mortality rates, among others [29]. In our study, *C. albicans* was detected in all groups. However, in one MG patient (A07), *C. albicans* was dominantly present. According to the Human Microbiome Project and their study on gut mycobiomes of healthy subjects, fungal communities were characterized by a high prevalence of yeast including *C. albicans*, with operational taxonomic units (OTUs) present in 80.8% of samples [30]. Although there was a high degree of inter-subject variability in fungal communities, *C. albicans* amplicon sequence variants (ASV) were found in 63.6% of subjects. In contrast to bacteria in the digestive tract, *C. albicans* colonizes all segments from the oral cavity to the anus [31]. We hypothesize that subject A07 might have had an intestinal fungal overgrowth or an oral candidiasis. Both often occur after long-term antibiotic treatment or in short bowel syndrome, a malabsorptive disorder, as a result of the loss of bowel mass mostly secondary to surgical resection of the small intestine. Both conditions were excluded by study inclusion criteria (no history of treatment with antibiotics within the previous 4 weeks and no short bowel syndrome). 

However, some fungal sequences could not be assigned to fungal genera or species, which reduced the impact of the mycobiome analysis. Therefore, comparison between groups was not reasonable based on the obtained data. To extract meaningful community profiles for fungi, challenges must be overcome [22] which are in parts different from those of bacteria, e.g., recognition of process-induced sequencing errors [32], accurate taxonomic assignments to define community members and structure [33] as well as binary naming and phylogenetic classifications [34]. In our study, limited content and annotation most likely led to non-robust fungal identification or failure to achieve a genus-level assignment. The applied DNA extraction protocol was appropriate for PCR amplification of fungal regions regarding DNA yield and quality [35]. However, the best extraction protocol for analysis of mycobiome data was achieved using the standardized IHMS Protocol Q with additional repeated beat-beating steps [36]. Furthermore, sequencing of the ITS2 region provides greater resolution of the relatively low-abundance mycobiome species in comparison to 18S rRNA gene sequencing [30]. In contrast to the former bacterial analysis, taxonomic classification and interpretation of fungal results was more challenging [37], although the UNITE database, a well-curated, high quality database that is constantly updated, was utilised [38]. To discard non-target sequences and to increase taxonomic resolution, Nilsson et al. recommend pipelines such as PipeCraft, PIPITS, LotuS and AMPtk for Illumina data without erasing all errors that occurred during sample preparation and sequencing [39]. The parallel use of different taxonomy assignment tools for comparison and combination of results may be helpful [40]. 

As a sub-analysis from the initial study performed between July 2017 and August 2018, DNA aliquots were stored at −80°C after DNA extraction and were twice thawed, namely once for the bacterial microbiome analysis and again for the present mycobiome analysis. This might have had an impact on the sequencing yield and quality but should be only marginal. 

In MS, the gut mycobiome differs from that of healthy individuals, with enrichment of *C.* and *Epicoccum* as well as a lower relative abundance of *Saccharomyces* [8] and over-representation of *Saccharomyces* and *Aspergillus*, respectively [7]. Different mycobiome profiles could be defined with different immune cell subsets in the blood [7]. Due to the above stated reasons, a distinct gut mycobiome signature could not be generated and, therefore, comparisons with other autoimmune neurological diseases such as MS are difficult. If compared to the recently described high prevalence of *Saccharomyces*, *Malassezia*, and *C.* within the gut mycobiome of the Human Microbiome Project healthy cohort, our findings within the healthy volunteer group HV bore a resemblance in part [30]. 

## 5. Conclusions

This study evaluated potential alterations of the gut mycobiome in patients with MG in comparison to subjects with other (autoimmune) neurological diseases and healthy subjects, and found no difference between the groups. However, due to the small groups, further investigations are required to assess associations of the mycobiome with MG and interactions between fungi and autoimmunity, including analysis of fungal functional profiles and immune responses as well as unravelling the significance of fungal species with acetylcholinesterase inhibitory activities in MG.

## Figures and Tables

**Figure 1 jof-09-00569-f001:**
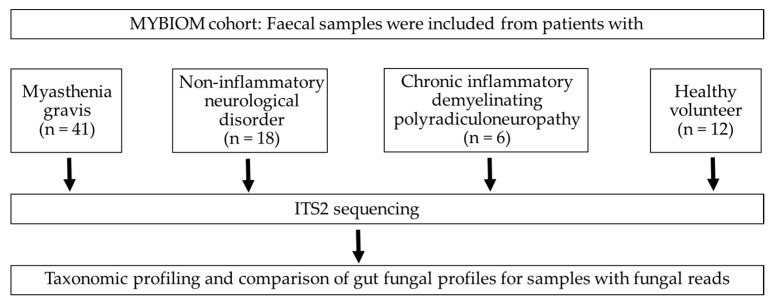
Flow chart and enrolled participants in the study.

**Figure 2 jof-09-00569-f002:**
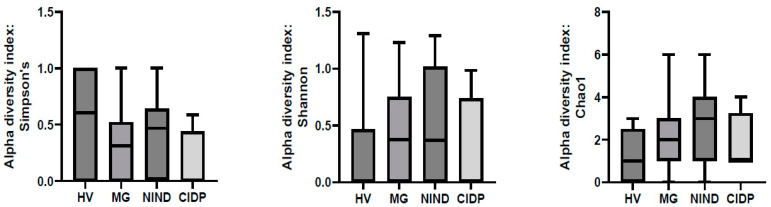
Comparison of alpha-diversity indices of the different study groups MG, NIND and CIDP using the Kruskal–Wallis test. Data are presented as mean ± standard error of the mean. HV, healthy volunteer; MG, myasthenia gravis; NIND, non-inflammatory neurological disorder; CIDP, chronic inflammatory demyelinating polyradiculoneuropathy.

**Figure 3 jof-09-00569-f003:**
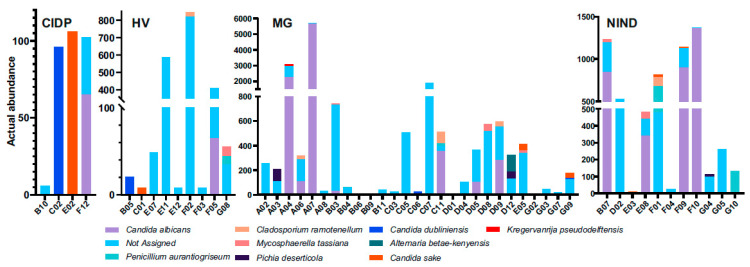
Differentially abundant fungal species in faecal samples from patients. HV, healthy volunteer; MG, myasthenia gravis; NIND, non-inflammatory neurological disorder; CIDP, chronic inflammatory demyelinating polyradiculoneuropathy.

## Data Availability

Sequenced reads generated in this study have been deposited in the European Nucleotide Archive (https://www.ebi.ac.uk/ena/browser/home) (accessed on 4 July 2022). under accession number PRJEB61825.

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
