# Peer review of "Fungal Gut Microbiome in Myasthenia Gravis: A Sub-Analysis of the MYBIOM Study"

_jof, 2023, doi:10.3390/jof9050569_

Round 1

Reviewer 1 Report

The manuscript of Verhasselt et al. presents the data on the fungal gut microbiome in myasthenia gravis patients. The data was derived from the MYBIOM study, a single-center study on gut microbiota. There was no prominent fungal dysbiosis in the MG group found, however, the major limitation is that a significant number of mycobiome species could not be assigned

Major comments:

1. A significant number of species could not be assigned-are there any other pipelines developed in the meantime that could solve this problem? How do other studies deal with that problem? Please describe even if the methods are not applied here.

2. Introduction-please develop more on the fungal microbiota in disease 

3. Are there any studies on mycobiome in the larger cohorts in MG?

Author Response

We thank the reviewer for the comments and hope that we have dealt with all points satisfactorily.

Major comments:

  1. A significant number of species could not be assigned-are there any other pipelines developed in the meantime that could solve this problem? How do other studies deal with that problem? Please describe even if the methods are not applied here.

We now included sentences in the discussion about how other working groups solved the problem with failed assignment or which pipelines etc they suggest. [lines 189-193]

  1. Introduction-please develop more on the fungal microbiota in disease. 

We are thankful for this point. We included more references to explain the impact on fungal microbiota in disease in more detail.  [lines 46-52]

  1. Are there any studies on mycobiome in the larger cohorts in MG?

We thank the reviewer for this comment. To the best of our knowledge, there are no larger cohorts investigating the mycobiome in MG available.

Reviewer 2 Report

In this manuscript, Fungal gut microbiome in Myasthenia gravis: A sub-analysis of 2 the MYBIOM study was studied well. But there are some questions in the aspects of experimental designs, results and discussion and so on.

Hence, I have some suggestions as follows:

1) Some descriptions in the manuscript were not exact or confusing. Some words which will make the manuscript feel like an article on a popular science book should not appear in such a research paper. The following are suggestions for improving English usage. Please use standard expression in English. 

2)The manuscript stays within a stage of literature survey, and is hard to find original contribution of the authors on this subject.

3) Problems on format or details: the manuscript was not well prepared according to the “Guidelines”. Please check carefully. The abstract should be added some quantitative data.

4)If you put some photos and schematic diagram into the paper, the design of your manuscript will be more clearly understood.

Author Response

We thank the reviewer for the comments and hope that we have dealt with all points satisfactorily.

  1. Some descriptions in the manuscript were not exact or confusing. Some words which will make the manuscript feel like an article on a popular science book should not appear in such a research paper. The following are suggestions for improving English usage. Please use standard expression in English. 

We thank the reviewer for this point and we changed single phrases throughout the manuscript and improved language.

  1. The manuscript stays within a stage of literature survey, and is hard to find original contribution of the authors on this subject.

At this point, we disagree with the reviewer. Our manuscript contains new and interesting data on a topic in the neuromuscular sector that has remained rather untouched until now. Since it is a sub-analysis of a previously conducted and published study on neuromuscular patients, the data is not abundant and, therefore, submitted as a communication and not as an original article. If we have not understood the reviewer correctly on this point, we ask for clarification from the editor, if necessary.

  1. Problems on format or details: the manuscript was not well prepared according to the “Guidelines”. Please check carefully. The abstract should be added some quantitative data.

We now adhere to the instructions for authors and added numbers to the abstract. [lines 22-24]

  1. If you put some photos and schematic diagram into the paper, the design of your manuscript will be more clearly understood.

We are thankful for this point. We now included a schematic diagram in the Materials and Methods  section. [lines 75-77]

Round 2

Reviewer 2 Report

The manuscript has been revised a lot and the paper can be publishd for more discussion in the field.

Author Response

To the reviewer,

Thank you very much. We did a spellcheck. We have addressed the comment and updated our Data availability statement by adding an accession number.

Kind regards,

Andreas Totzeck, MD
